# Which Epidemiological Characteristics Drive Decision Making in the Management of Patients with Vestibular Schwannoma?

**DOI:** 10.3390/biomedicines11020340

**Published:** 2023-01-25

**Authors:** Zdeněk Fík, Aleš Vlasák, Eduard Zvěřina, Jaroslav Sýba, Jan Lazák, Lenka Peterková, Vladimír Koucký, Jan Betka

**Affiliations:** 1Department of Otorhinolaryngology and Head and Neck Surgery, 1st Faculty of Medicine, Charles University and Motol University Hospital, 15006 Prague, Czech Republic; 2Department of Neurosurgery, 2nd Faculty of Medicine, Charles University and Motol University Hospital, 15006 Prague, Czech Republic; 3Department of Otorhinolaryngology, 2nd Faculty of Medicine, Charles University and University Hospital Královské Vinohrady, 10034 Prague, Czech Republic

**Keywords:** vestibular schwannoma, observation, tumor growth, surgery

## Abstract

The incidence of sporadic vestibular schwannoma has significantly increased over the past few decades. However, there is no method currently available to accurately predict the risk of subsequent tumor growth. The difference in the management of five patient groups has been evaluated: wait and scan, conversion to microsurgery, conversion to stereoradiotherapy, sterioradiotherapy, and microsurgery. In total, 463 patients with vestibular schwannoma have been consulted in our department from 2010 through 2016. Of the 250 patients initially indicated for observation, 32.4% were later indicated for active treatment. Younger patients were more frequently indicated for surgery (mean age 48 years) compared to older patients, who were more often indicated for stereoradiotherapy (mean age 62 years). Tumor growth was observed more often in patients under 60 years of age and in patients with tumors greater than 10 mm. In elderly patients, including those with larger tumors, a conservative approach is the optimal solution. If tumor growth occurs in the wait-and-scan strategy, it is still possible to continue with a conservative approach in some situations. The duration of follow-up scans is still a matter of debate, as tumors can begin to grow after 5 years from the initial diagnosis.

## 1. Introduction

One of the most important advances in vestibular schwannoma (VS) treatment is the increasing number of patients indicated for observation after initial diagnosis. Few articles describe optimistic outcomes in patients with large tumors (Koos IV) managed with conservative strategies [1]. However, the improvement in CPA surgery decreased complication rates to levels in which small tumors can be indicated for active treatment with the aim of preserving neural function [2].

The previously mentioned increasing significance of the conservative approach in the management of patients with VS is attributable to the increased diagnostic sensitivity of magnetic resonance imaging, together with improved awareness of the diagnosis itself [3]. Furthermore, there has been an increase in the incidence of VS in the population, shifting from 1–2/100,000 to nearly 5/100,000 in some studies [4,5].

The crucial pattern of the tumor is its biological behavior. Approximately 30–70% of VS exhibit growth; conversely, 30–70% remain stable, leaving a small percentage of VS that shrink over time (5–10%) [6,7]. The average growth rate ranges between 1 and 2 mm/year. Faster growth rates (more than 5 mm/year) are rare and indicate a potential failure of radiotherapy [8,9].

VS growth activity varies over decades of life. The growth activity of VS in elderly patients is less progressive compared to younger patients. This knowledge of growth activity differences in VS in elderly patients allows for their observation, including those with larger tumors in size. It raises the question of whether to indicate stereoradiosurgery in these age groups [10].

Understanding the biological behavior of a tumor could help us stratify patients into appropriate treatment groups, achieving the most satisfactory results.

## 2. Materials and Methods

Data collection was retrospectively obtained in the years 2010–2016, including all patients with a morphological diagnosis (MRI—magnetic resonance imaging) of VS in the Department of Otorhinolaryngology and Head and Neck Surgery, First Faculty of Medicine, Faculty Hospital Motol. Data were obtained via telephone calls.

In surgically treated patients, a histological examination was performed, confirming the diagnosis of vestibular schwannoma. In solely observed or irradiated patients, we relied on MRI characteristics, proved by experienced radiologists. Vestibular schwannoma is isointense (or hypointense) on T1-weighted images and hyperintense on T2-weighted images. The gold standard is T1-Gadolinum enhancement, which should be intense. Regarding other possible CPA pathologies, the main concern is related to meningioma. Meningioma in the CPA is usually placed asymmetrically with respect to the position of the internal acoustic meatus, and it is less likely to grow into the meatus. Moreover, meningioma is more homogenous and less enhancing the Gadolinum. Finally, a dural tail (thickening of the adjacent dura) is a common sign of meningioma.

The patients were divided into five groups according to the chosen treatment.

(a)(wait and scan):
-Includes all patients indicated for observation who remained untreated until 12/2021.-Follow-up consists of repeated magnetic resonance imaging (MRI), usually 6 months after initial diagnosis and then once annually for at least 3 years. The subsequent lengthening of the interval between follow-ups depends on the morphological and functional stability of the tumor, the age, and the preferences of the patients.(b)SRT (stereoradiotherapy):
-Includes patients indicated for stereotactic irradiation mainly, but not exclusively, using the Leksell Gamma knife (LGN). The interval between initial diagnosis and treatment should be less than 6 months.(c)Surgery:
-Includes patients indicated for surgery with an interval between the initial diagnosis and treatment of less than 6 months.(d)WaS—Surgery:
-Includes patients indicated for observation who were later indicated for surgery. The minimum interval between initial diagnosis and surgery was 6 months, with at least one control MRI performed during this interval.(e)WaS—SRT:
-Includes patients indicated for observation who were later irradiated. The minimum interval between diagnosis and surgery was 6 months, with at least one control MRI performed.

The basic parameters monitored in these groups of patients were sex, age, tumor size according to Koos and international classification [11], tumor growth, and the interval between initial diagnosis and conversion to active treatment.

Generally, patients with smaller tumors (Koos II and II) or older patients with larger tumors (Koos III and IV) are indicated for conservative management. Surgery is initially considered in patients with larger tumors (Koos III and IV). Stereoradiosurgery is indicated in patients with growing tumors, not exceeding 2.5 cm in the largest diameter. The patients themselves play crucial roles in the treatment indication.

Descriptive and inductive statistic methods were used for statistical analysis. Among the methods of descriptive statistics, the methods of moment involved sample means and sample standard deviations, and the quantile methods involved medians. Data normality was verified using the Shapiro–Wilk test, Anderson–Darling test, and Kolmogorov–Smirnov test. Among the methods of inductive statistics, the Wilcoxon rank sum test, two-sample *t*-test, analysis of variance, Bravais–Pearson correlation coefficient (including the test of its statistical significance), Spearman correlation coefficient, and the chi-square test of goodness of fit with correction for continuity according to Yates (including Haberman residuals).

## 3. Results

A total of 463 patients (254 women and 209 men) were consulted in our department during the designated period. The mean age was 54 years (12 to 86 years). In total, 137 tumors were initially Koos I, 114 were Koos II, 70 were Koos III, and 142 were Koos IV. According to the INT classification, 137 patients were in stage 0, 103 were in stage 1, 98 were in stage 2, 58 were in stage 3, 31 were in stage 4, and 9 were in stage 5. In 27 patients, we were unable to find the data due to the retrospective design of the study (Table 1). The average diameter of the tumor was 17 mm (1–50 mm).

In the 0–59 y group, 54% were women and 46% were men, compared to 56% being women and 44% being men in the 60+ group (*p* = 0.485).

The average duration of follow-up was 46 months (max 125 months). A total of 51 patients visited our department only once, and in most of these cases, the data were collected via telephone calls. The rest of the data were collected with the collaboration of other ENT and neurosurgical departments. Thanks to this careful work, there no patients dropped out of the study.

The conservative strategy was indicated in 250 patients (54%); however, 54 patients were later indicated for surgery and 27 for SRT (Figure 1). Therefore, in general, 32% of patients were converted to active treatment. The remaining 169 patients (36.5%) remained in the WaS group. In the WaS group, most tumors were stable, and in 15 cases (9%), growth/shrinkage in tumor size was observed in the monitored period. The size of the growth ranged from 1 to 10 mm, and in six patients, the growth led to an increase of one point in the Koos classification. Tumor shrinkage was observed in 3 patients (3–6 mm; 1.8%), resulting in an improvement in the Koos classification in one case. In general, 30% of the patients, mostly those initially indicated for observation, experienced tumor growth. In total, 19 patients died during the designated period.

If stratified according to the Koos classification, in Koos I, 95.6 % of the patients were initially indicated for WaS, and the remaining patients were indicated for surgery (3.7%) and SRT, respectively (0.7%). At the end of the study, 25.6% of the patients in the WaS group were converted to active management, creating a final ratio of 71%:8%:21% for WaS:SRT:surgery. However, only 6% of the patients with Koos IV tumors were initially observed, and 25% of them had to be subsequently indicated for active treatment. None of these observed patients were diagnosed with hydrocephalus. The active treatment of patients with Koos IV tumors mainly consisted of surgical resection—86.6% initially and 88% in total, accounting for patients with the initial indication of surgical resection and patients converted to surgical resection at the end of the study. For more detailed results of the treatment stratification, see Figure 2.

Regarding the ratio of patients converted to active therapy, it gradually increased together with the initial Koos grade. In the group of Koos I patients, 25.6% of the patients were re-evaluated and indicated for active treatment. In the Koos II group, 37.2% of patients were indicated for active treatment. In total, 48.3% of patients in the Koos III group failed to continue with the conservative strategy. On the contrary, Koos IV patients were reindicated in 25.0% of cases.

The mean interval from the follow-up to conversion to surgery was 18.5 months (5–97 months). There were two patients with non-growing tumors; the rest experienced an average growth of 3.7 mm (1–11 mm), which resulted in a change in the Koos classification in 30 cases (68%; data not available for 10 cases).

In 17 cases (31%), a change in indication was made immediately after the first MRI control. In eight cases, late onset of growth was observed, and the indication for surgery came later than 3 years after the first MRI. In four cases, the tumor exhibited slow growth over the course of years, and the indication for surgery came with the patient’s decision. In three cases, delayed growth was observed (start of growth 2 yars, 4 years, and 6 years after the initial diagnosis). In one case, there was a 9-year interval in which the tumor displayed shrinkage on MR images (observed at a different department), followed by slow tumor growth in the subsequent 3 years.

In 11 cases (20%), we did not know the reason for the change in treatment strategy since the patients were indicated for surgery and operated on in a different department. Interestingly, three of these extramurally operated-on patients were initially observed in our department for a duration of more than 3 years, with tumors showing no growth.

The mean duration of follow-up to conversion to SRT was 19 months (8 to 51 months). This group was exclusively treated using the LGN, except for one case in which the patient was treated using Cyberknife. There were three patients with non-growing tumors. The rest experienced an average growth of 2.8 mm (2–7 mm), which resulted in a change in the Koos classification in eight cases (42%; data not available for eight cases).

In six cases (22%), a change in the indication was made immediately after the first MRI control. In two patients (7%), stereoradiotherapy was indicated later than 3 years after the initial diagnosis and, in both cases, after the continuous growth of the tumor.

When we look at the two age groups of the patients, in the 0–59-year group, the mean time from the initial diagnosis to conversion to surgery or stereoradiotherapy was 22.6 and 13.2 months, respectively (*p* = 0.03162). On the contrary, in the group of patients 60 years and older, the waiting period for surgery was shorter (16.3 months) compared to that of stereoradiotherapy (19.8 months; *p* = 0.02419). In general, younger patients were indicated for conversion earlier from the time of initial diagnosis compared to the 60+ group (17.8 months vs. 19 months; *p* < 0.001).

In total, 23 patients (5%) were indicated for stereoradiotherapy during the first appointment. One patient died before the start of treatment. Most patients underwent irradiation with LGN (91%). The remaining two patients chose, of their free will, Proton beam therapy and LINAC, respectively.

Overall, 50 patients were treated with stereoradiotherapy. In six of them, recurrence was proven (12%; five patients underwent LGN, and one underwent Proton beam therapy), and revision surgery (four patients) or reirradiation (one patient) was indicated. In one patient, we could not find additional information.

In total, 190 patients (41%) were indicated for surgery during the first appointment. Three patients died before the start of treatment. Of the 190 patients, 181 were operated on in our department, and 6 were operated on elsewhere.

Looking at the individual treatment groups in relationship to gender, no correlation was found (*p* = 0.4361). Age appears to be a strong predictor of the treatment chosen (*p* < 0.001); younger patients were more frequently indicated for surgery (mean age 48 years). Conversely, older people were more frequently indicated for stereoradiotherapy (mean age 62 years). For complete results, see the table (Table 2). The size of the tumor also differed significantly between the groups (*p* = 0.004228), leaving larger tumors (average Koos 3.45; average largest diameter 24.6 mm) for immediate surgical intervention, and conversely, smaller tumors (average Koos 1.6; average largest diameter 10.96 mm) for the conservative strategy (Table 3).

Regarding tumor growth (considering both the Koos classification and the progression in millimeters), there were no significant associations between age, gender, and initial tumor size. However, it must be noted that we observed some trends, in which there was a greater representation of growing tumors in the younger patient group (0–59 y; 36% vs. 26%; *p* = 0.4565) and in the group of patients with tumors of initial sizes larger than 10 mm (35% vs. 25%; *p* = 0.2822) (Table 4).

The mean growth rate was 2.4 mm/year, the lowest was 0.7 mm/year, and the fastest growth rate seen was 15 mm/year due to cystic degeneration. When comparing tumors according to size, no differences were found between groups of 0–10 mm (mean 2.3 mm/year) and >10 mm (mean 2.4 mm/year; *p* = 0.5556). In women, we observed slightly faster growth rates (2.7 mm/year) compared to males (2.0 mm/year). This observation, however, did not reach statistical significance (*p* = 0.06457).

Generally, in the group of patients older than 60 years, there were smaller tumors (mean Koos 2.3) compared to the younger group (mean Koos 2.5). Although statistically significant (*p* = 0.01902), clinically, this was not strongly different (Table 5). On the other hand, Koos I tumors were more common in the group of older patients (34% vs. 27%) compared to Koos IV tumors, which were more common in the group of younger patients (35% vs. 23%; *p* = 0.0138).

## 4. Discussion

The wait-and-scan strategy has become a widely accepted approach for patients with vestibular schwannoma, as it is believed that most tumors do not grow after diagnosis. When looking at the group of patients initially approached with the conservative strategy, only 30% of patients experienced an increase in tumor size, and in 83% of patients within this group, tumor growth led to conversion to active treatment. The average follow-up time was 3.8 years. These data are similar to another observational study, in which 32% of the patients had a growing tumor during the mean follow-up time of 3.6 years [12].

In general, in 32% of the patients, the initial WaS strategy had to be re-evaluated, with later indication for the active approach. In the literature, the rate of conservative management failure differs significantly, from 19% to 56% [13,14,15]. If stratified according to tumor size, with the increasing size up to Koos III, the probability of the failure of the WaS strategy increased to 48%. Interestingly, the rate of re-evaluated patients with Koos IV tumors was only 25%, which was less than that of patients with Koos I tumors. The reason for this could lie in the decision to solely observe patients with comorbidities, advanced age, or due to their preference, despite them having large tumors, as the risk of taking an active approach in these patients could outweigh the potential benefit. However, of the nine observed patients with Koos IV tumors, seven were stable, one had a small 1 mm progression, and one patient experienced shrinkage.

In patients indicated for the wait-and-scan strategy, active treatment was offered in the case of tumor growth; however, several patients converted to active treatment even in the case of a nongrowing tumor. In the surgical group, overall, two patients were converted from the conservative strategy; one case was due to the wish of the patient, and the other due to progression in hearing loss, in an attempt to preserve hearing function.

Three patients in the SRT group underwent the procedure on their own initiative despite our recommendation to continue with observation.

In the WaS group, 12 patients who experienced some range of growth were not indicated for active treatment. The main reasons were age and comorbidities (eight patients) and the refusal of active intervention (four patients). In five of these patients (42%), the tumor stopped growing.

When looking at the WaS group, regardless of whether the latter converted or not, subsequent growth was observed in 30% of cases. The probability of growth appeared to be higher for larger tumors extending beyond the meatus [16,17]. There was also a trend in our study to observe a higher growth rate of tumors initially exceeding 10 mm in their largest diameter; however, this could not be statistically proven. In the same way, younger age should be a negative prognostic factor for tumor stability [18], which was not significantly observed in our study; a higher growth probability was found in patients younger than 60 years. On the other hand, this finding is inconsistent, and some of the studies failed to prove this [13,14]. In either case, younger patients were more frequently indicated for active treatment in our study and were indicated earlier for conversion compared to older patients.

Tumor growth is more likely to occur in the 5-year period after the initial diagnosis [16,17]. However, a later onset of growth is possible; therefore, it is not reasonable to stop scanning patients. The only question remaining is with regard to elderly patients [13,14]. Borsetto et al. described a maximum growth interval of 42 months in patients older than 70 years [19]. In this group of patients, lifelong surveillance can be discussed.

## 5. Conclusions

Tumor size is a strong predictor of the treatment modality chosen; however, in elderly patients, the conservative approach is an optimal solution, including those with larger tumors, as there is an increased probability of non-growing ability.

If growth occurs in the wait-and-scan strategy, in some situations, it is still possible to continue with the conservative strategy since some tumors exhibit no further growth. However, there is no method currently available to accurately predict VS behavior.

As tumor growth can occur even more than 5 years after the period of growth stability, lifelong surveillance using MRI is a matter of debate.

## Figures and Tables

**Figure 1 biomedicines-11-00340-f001:**
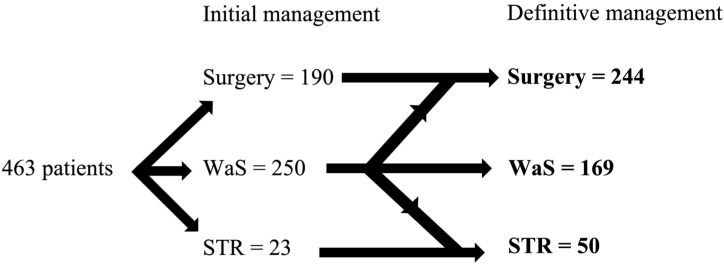
Distribution of patients into different treatment groups. WaS—wait and scan, STR—stereoradiotherapy.

**Figure 2 biomedicines-11-00340-f002:**
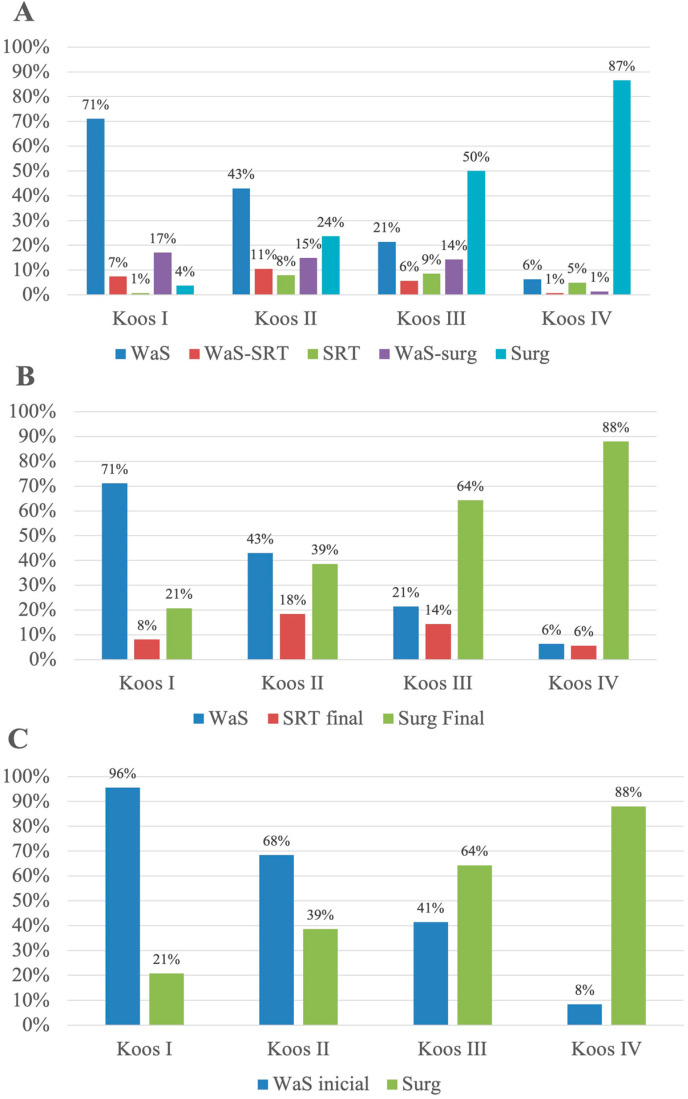
Distribution of patients into treatment groups according to the tumor size. (**A**) Including conversions, (**B**) final situation, and (**C**) difference between initial WaS and surgery indications.

**Table 1 biomedicines-11-00340-t001:** Size of the tumor according to Koos and international classification.

Koos	*n*	INT	*n*
I	137	0	137
II	114	1	103
III	70	2	98
IV	142	3	58
		4	31
		5	9
		NA	27

**Table 2 biomedicines-11-00340-t002:** Mean age in the treatment groups with the statistical comparison.

Modality		WaS–STR	STR	WaS–Surgery	Surgery	WaS
	**Age**	60.2	62.8	47.8	48.3	59.4
**WaS–STR**	60.2		>0.05	<0.001	<0.001	>0.05
**STR**	62.8	>0.05		<0.001	<0.001	>0.05
**WaS–surgery**	47.8	<0.001	<0.001		>0.05	<0.001
**Surgery**	48.3	<0.001	<0.001	>0.05		<0.001
**WaS**	59.4	>0.05	>0.05	<0.001	<0.001	

**Table 3 biomedicines-11-00340-t003:** Mean size of the tumor in the treatment groups with the statistical comparison.

Modality		WaS–STR	STR	WaS–Surgery	Surgery	WaS
	**Tumor size (mm)**	12.6	18.1	12.0	24.6	11.0
**WaS–STR**	12.6		>0.05	>0.05	<0.001	>0.05
**STR**	18.1	>0.05		<0.05	<0.01	<0.05
**WaS–surgery**	12.0	>0.05	<0.05		<0.001	>0.05
**Surgery**	24.6	<0.001	<0.01	<0.001		<0.001
**WaS**	11.0	>0.05	<0.05	>0.05	<0.001	

**Table 4 biomedicines-11-00340-t004:** Tumor growth with the respect to the age, sex, and size of the tumor.

	Age		Gender		Initial Size	
	0–59	60+	F	M	0–10 mm	11+ mm
Average growth (mm)	1.3	0.90	1.1	1.1	0.98	1.2
% of growing	36%	26%	32%	29%	25%	35%
Average growth (mm)	3.7	3.5	3.4	3.8	4.0	3.4

**Table 5 biomedicines-11-00340-t005:** Tumor size according to the age groups of patients.

	0–59	60+
**Koos 1**	27%	34%
**Koos 2**	25%	24%
**Koos 3**	14%	17%
**Koos 4**	35%	23%

## Data Availability

The data presented in this study are available on request from the corresponding author.

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
