# Peer review of "Which Epidemiological Characteristics Drive Decision Making in the Management of Patients with Vestibular Schwannoma?"

_biomedicines, 2023, doi:10.3390/biomedicines11020340_

Round 1
Reviewer 1 Report
The authors evaluated the impact of epidemiological characteristics of patients harboring vestibular schwannomas, on the therapeutic strategy. In fact, they analyzed a retrospective cohort of 463 patients (from 2010 to 2016) to assess if age and other characteristics drive to a wait-and-see strategy, a first-line surgical indication, or a gamma knife treatment.
The management of VS still remains controversial, in particular in the case of first diagnosis in asymptomatic patients and incidental tumors. The authors assessed that in general, younger people are more prone to faster conversion into active treatment compared to older ones, and the size of the tumor is a strong predictor of the treatment modality chosen; however, in elderly patients, it appears that the conservative approach is an optimal solution even in larger tumors.
My main question about the manuscript is related to the criteria leading to the first treatment, which should be more clearly stated. Moreover, it should be indicated the management of hydrocephalus in huge tumors mainly in elderly patients: sometimes, a conservative management on the tumors itself does not automatically mean lack of invasive treatment, in particular in the case of an older population. Therefore, this should be indicated
Author Response
Dear reviewer,
thank you very much for positive and constructive comment. I have added information about our selecting criteria, however find it difficult to clearly state some unsurpassable criteria for distinct group of patients. I always mention, there are 10 factors, affecting the treatment chosen and at the end, patient is the main one to decide.
Regarding the hydrocephalus, I looked only on patients with large vestibular schwannomas in WaS strategy. In those patients, surprisingly, we were not dealing with hydrocephalus, even In patient with 43mm tumor.
All changes in the text are marked in yellow.
Reviewer 2 Report
Dear authors,
Thank you for submitting your research under consideration for publication in Biomedicines. The manuscript biomedicines-2139850 is a retrospective cross-sectional clinical study on the epidemiologic characteristics influencing the therapeutic strategy for the vestibulary Schwannoma. The authors conclude that the primary tumor size and age of the patient are essential determining factors between active surveillance, surgery, and stereotactic (gamma-knife) radiosurgery.
The manuscript has the strength of reporting on a large patient database (N>430) using proper statistical methods and adequate graphs. As an overall disadvantage, the study comes from a single center and lacks novelty. To support the merit for publication, authors are advised to reconsider their manuscript according to the following peer suggestions:
MAJOR COMMENTS
C001_The manuscript will fundamentally benefit from professional language proof editing with medical terminology proficiency. A receipt of the official service or a statement of the assigned native speaker shall be submitted with the corrected manuscript for the interest of the editor.
C002_Study design: Please clarify the following points:
i) What was the gold standard for the diagnosis of Schwannoma in patients that did not undergo surgery? How did you exclude alternative clinical entities with similar morphology (e.g. meningioma) but different progress curves?
ii) Kindly report on the inclusion and exclusion criteria
iii) Kindly report on statistical handling with drop-offs and deaths during the study
C003_Related to C002_i: The study population was overall sex- and stage-balanced. The readers might wonder whether the balanced profile of the study still applies to different age groups. The report would benefit a lot from more detailed descriptive statistics, including the tumor size and sex distribution for younger and older patients.
A special group of interest is the group of premenopausal women and whether hormone dependency might influence the overall outcome of active surveillance -vs- gamma knife – vs- surgery
C004_Could the authors potentially report the % of patients that were converted to surgery after relapse of gamma-knife treated tumors?
Author Response
Dear reviewer,
thank you for constructive comments. Please, find my responses below (in the text, all changes marked in yellow, except for English proofreading).
C01: The text went through proofreading
C02: i) Thank you for this comment. I agree that this needs to be explained. I have added more information in the Material and Methods.
ii) The only inclusion criteria was the diagnosis of vestibular schwannoma. There was no exclusion criteria with the respect to the point "C02 i".
iii) I have added information about deaths in WaS group. Those patients did not affect the aim to focus on the primary decision making. Moreover, thanks to the collaboration with other ENT and NS departments, together with phone calls, we were able to collect data retrospectively from all initially lost patients.
C03: More information added together with the new Tab 5. The note about analysis of woman with the respect to their hormonal status is very interesting, but unfortunately it was impossible to complete this statistic.
C04: information added
Round 2
Reviewer 1 Report
The manuscript could be suitable for publication in the current, revised form
Reviewer 2 Report
Dear authors,
Dear Editor,
Thank you for overworking the manuscript over the holiday season. The revised manuscript version fulfills all reviewer´s comments and fits the aims and purposes of your Journal.
Authors should make their database publicly available for deep learning training issues.
With best regards and all jolly greetings for the new year,